# A Novel Force Variation Fine-Blanking Process for the High-Strength and Low-Plasticity Material

Huajie Mao [1,2,3], Han Chen [1,2,3], Yanxiong Liu [2,3,4,*] and Kaisheng Ji [2,3]

1    School of Materials Science and Engineering, Wuhan University of Technology, Wuhan 430070, China; maohj@whut.edu.cn (H.M.); chwhut@whut.edu.cn (H.C.)
2    Hubei Key Laboratory of Advanced Technology for Automotive Components,
     Wuhan University of Technology, Wuhan 430070, China; jiks@whut.edu.cn
3    Hubei Key Collaborative Innovation Center for Automotive Components Technology,
     Wuhan University of Technology, Wuhan 430070, China
4    Hubei Engineering Research Center for Green Precision Material Forming, Wuhan University of Technology,
     Wuhan 430070, China
*    Correspondence: liuyx@whut.edu.cn

**Abstract:** High forming force is often needed when high-strength and low-plasticity materials are processed by fine blanking. Too high forming force increases the load of the die and greatly increases the risk of die failure. If the forming force is reduced, the material will fracture prematurely, which will lead to poor quality parts. Aiming at this problem, a new force variation load fine-blanking technology is proposed in this paper. During the loading process, the forming force does not remain constant but changes with the blanking stroke. A 2D finite element fine-blanking model was established for the TC4 material. The mechanism of force variation fine blanking is also revealed. This paper proposes a method to design the loading route of the forming force with variable load. This method combines finite element simulation with neural network and a multi-objective genetic algorithm. Finally, the application of variable load fine blanking production and the application of traditional fine blanking production parts are verified by the experimental method. The same results are obtained from both simulation and experiment. It is found that the variable load fine blanking process can greatly reduce the load of the die on the premise of ensuring the quality of fine-blanking parts.

**Keywords:** fine blanking; force variation; neural network; genetic algorithm

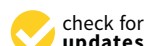



## 1. Introduction

Fine blanking is an advanced sheet metal plastic-forming process that is derived from the conventional blanking process. In the fine-blanking process, only one punching stroke is required to obtain parts with high surface quality and dimensional accuracy. Compared with conventional blanking, the efficiency is greatly improved, and the manufacture cost in the production process is reduced. The dimensional accuracy of fine-blanked parts can reach the IT7 level, and the surface roughness of the shear surface can reach *Ra* 0.3 at the highest. Currently, it is widely used in aerospace, automobile, shipbuilding, and other manufacturing fields [1].

To gain a high-quality cutting surface of fine-blanking parts, the key is that the material in the shear deformation zone undergoes plastic deformation as much as possible and delays the occurrence of fracture behavior [2]. For this reason, it is necessary to ensure a high hydrostatic stress state inside the material. Accordingly, fine-blanking clearance is generally only about 1% of the sheet thickness, which is far less than conventional blanking. This will be beneficial to the maintenance of the hydrostatic stress state inside the material. In addition, it also depends on the blank holder and the counter punch in the fine-blanking die structure. The V-ring on the blank holder can largely limit the outward

flow of internal materials. The layout of the V-ring needs to consider the thickness of the sheet. At present, the thickness of the parts produced by fine blanking is mainly less than 8 mm, and the maximum thickness of the parts can reach more than 20 mm. For thick parts, even double-sided V-rings are needed to clamp the parts. It cooperates with the counter punch to improve the hydrostatic stress and plastic-forming ability of the material in the shear deformation zone [3] and finally obtain high-quality parts [4]. For thick parts, even double-sided V-rings are needed to clamp the parts. However, the application of fine blanking also has certain disadvantages. Compared with ordinary blanking, fine blanking has a larger number of dies, and at the same time, fine blanking requires higher precision of the dies. Therefore, the manufacturing and maintenance costs of fine-blanking molds are relatively high. This leads to a certain increase in the manufacturing cost of the parts.

Based on the above basic principles, due to severe plastic deformation of the material during the processing, the material is required to possess good plasticity. Therefore, some materials need to be spheroidized and annealed before fine blanking to improve their plasticity [5]. According to this, Zheng et al. [6] used heat-assisted fine blanking to process the parts. This will potentially be applied to high-strength materials. The materials of parts often pursue higher specific strength during selection, when lightweight is becoming a hot topic nowadays [7]. As a representative of these materials, titanium alloys have excellent mechanical properties and are extensively used. Applying fine blanking can improve the overall performance of titanium parts and expand the application range of titanium alloys. However, their plasticity is poor, which makes them prone to fracture during the plastic process [8].

At present, there are some studies on the fine blanking of high-strength materials. Gram et al. [9] and Sen [10] verified the possibility of fine blanking of high-strength materials through the experimental study of various strength materials. Zhao et al. [11] studied the behavior of DP600 high-strength steel during fine blanking and discussed the role of shear damage in induced tearing. Although the above research content involves the fine blanking of high-strength materials, it does not provide a solution to the problem of cracks during the forming process. If the forming force of high-strength material is calculated according to the empirical formula [12], the forming force value will be very large. According to the theoretical model proposed by Yin et al. [13], die wear is positively correlated with blank holder force and counter punch force. Therefore, although high forming force can improve the quality of parts, it can also dramatically increase the load of the mold. This will greatly affect the life of the mold and improve the use cost of the material, which is the main problem of fine blanking for high-strength materials now.

When encountering this problem, most researchers only explain the difficulties of fine blanking of high-strength materials, or alleviated the problem from the perspective of mold improvement [14]. Fine blanking of high-strength steels leads to an increase in the wear of fine-blanking punches. Klocke et al. [15] found that deep rolling has a potential to improve the wear resistance of fine blanking punches when processing high-strength material, and a novel profiled deep rolling tool is developed in this work. Bear et al. [16] performed a comprehensive numerical study on the influence of process parameters. It allows for prediction of the punch load during fine blanking of high-strength steels.

Based on the above problem, this paper proposed a new force variation fine-blanking process to form high-strength materials. It means that the value of the forming force is not constant during the blanking process. They continuously change according to the blanking stroke. It is found that the load on die always increases first and then decreases in the process of fine blanking. It keeps high load in the early stage of the blanking stroke and gradually decreases in the middle and late stage. However, the parts in the process of processing the demand for forming force are gradually increasing. Therefore, in the process proposed in this paper, the forming force keeps a low value in the early stage and a high value in the middle and late stage. Their trends are different. In this way, the load on the mold can be greatly reduced under the premise of ensuring the quality of parts.

In this paper, a 2D finite element simulation model for fine blanking was established. The influence of forming force on parts was explored, and the mechanism of force variation fine blanking was proposed. The forming force generally includes three parts: namely, the main blanking force, blank holder force, and counter punch force. However, in the fine-blanking process, the main blanking force is imposed by the press to make the part to be deformed and cannot be regulated accurately. Therefore, the control of forming force in this paper mainly focuses on blank holder force and counter punch force. Based on a large amount of simulation data, a neural network model was established with two forming force values as input and the length of the clean cutting surface as output. The genetic algorithm was used to optimize the neural network model. The die load and the length of the clean cutting surface are set as the two optimization goals for multi-objective optimization. The purpose of this paper is to improve the quality of parts by means of force variation load while ensuring the same mold load—or, on the premise of ensuring the quality of parts, greatly reducing the load of the mold. This will help to reduce the manufacturing cost of high-strength and low-plasticity materials and broaden their application.

## 2. Fine-Blanking Finite Element Simulation

### 2.1. Material Model

Titanium alloy is a high-strength material with a wide range of applications [17]. Therefore, TC4 titanium alloys were selected as the material in the simulation in this paper. The main components of the materials used in this paper are shown in Table 1.

**Table 1.** Main chemical constituents of TC4 titanium alloy.

| Al | V | Fe | O | C | N | H | Ti |
|---|---|---|---|---|---|---|---|
| 5.50 | 3.50 | 0.30 | 0.09 | 0.09 | 0.04 | 0.01 | others |

In this simulation, the J-C model [18] was selected as the material model, and the Oyane criterion was selected as the material fracture criterion [19]. The J-C model is shown in Equation (1). Due to the small temperature change in the process of fine blanking, the part related to temperature in the model is omitted. The tensile sample is shown in a small diagram in Figure 1a. All tensile tests were carried out at 20 °C with loading rates of 1.5 mm/min, 30 mm/min, and 60 mm/min, respectively, to simulate different strain rates. Each group was repeated three times. After the results of tensile tests are obtained, the parameters of the J-C model are fitted. Other material parameters obtained from the tensile test are placed in Table 2.

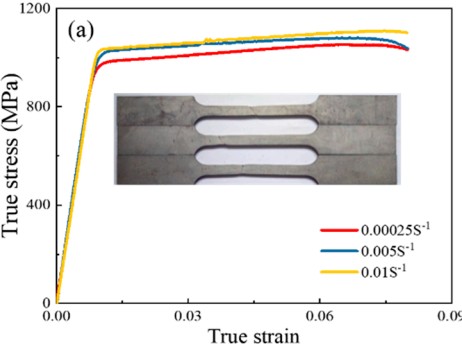
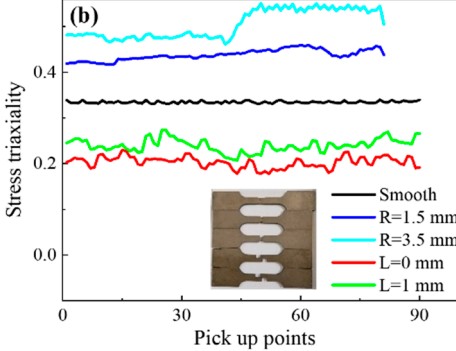

**Figure 1.** The sample and result curve used to establish the material model: (**a**) the tensile test sample diagram and the true stress–strain curve; (**b**) the Oyane criterion tensile test sample and the stress triaxiality value of the sample.

**Table 2.** Results of tensile test at room temperature.

| Strain Rate | $0.00025s^{-1}$ | $0.005s^{-1}$ | $0.01s^{-1}$ |
|---|---|---|---|
| Elastic modulus (GPa) | 113.5 | 112.3 | 112.5 |
| Poisson ratio | 0.31 | 0.31 | 0.31 |
| Yield strength (MPa) | 969.8 | 976.3 | 993.2 |
| Tensile strength (MPa) | 991.3 | 1019.5 | 1032.8 |
| Elongation at fracture (%) | 14.4 | 13.3 | 13.5 |

In Equation (1), $A$ is the yield stress value under initial conditions; $B$ and $n$ are parameters related to strain strengthening. $C$ is the constant representing strain rate sensitivity; $\dot{\varepsilon}_0$ represents the reference strain rate. $\sigma$ and $\varepsilon$ represent the stress and strain in the constitutive equation. $\dot{\varepsilon}$ is the strain rate.

$$\sigma = (A + B\varepsilon^n)\left(1 + Cln\frac{\dot{\varepsilon}}{\dot{\varepsilon}_0}\right) \tag{1}$$

Oyane criterion considers the influence of hydrostatic stress on material fracture, which is in line with the characteristics of fine blanking using hydrostatic stress to improve material fracture resistance. So, the Oyane fracture criterion is widely used in fine blanking. According to our previous work, Oyane criterion has good prediction accuracy for the process of fine blanking [20]. Oyane criterion is shown in Equation (2). The formula of stress triaxiality is shown in Equation (3).

$$C = \int_0^{\bar{\varepsilon}_f} (1 + A\eta)d\bar{\varepsilon}_p \tag{2}$$

$$\eta = \frac{\sigma_m}{\bar{\sigma}} \tag{3}$$

In Equation (2), $C$ is the fracture threshold; $\bar{\varepsilon}_f$ is the equivalent fracture strain; $\bar{\varepsilon}_p$ is the equivalent strain; $A$ is the material constant, representing the sensitivity of different materials to stress triaxiality; $\eta$ is the triaxiality of stress. In Equation (3), $\bar{\sigma}$ is the equivalent stress, and $\sigma_m$ represents the hydrostatic stress.

The parameters of the Oyane criterion were obtained by tensile test and finite element simulation. The shapes of tensile samples are different, as shown in Figure 1b, to simulate the equivalent fracture strain under different stress triaxiality. First, the tensile length of fracture was obtained through a tensile test. All experiments were repeated three times. Then, the finite element model is established to simulate the tensile process. When the tensile amount reaches the tensile length in the experiment, the equivalent strain at the center of the sample is read as the equivalent fracture strain. The point tracking function was used to make statistics on the stress triaxiality in the center of the sample, and the curve shown in Figure 1b was obtained [21]. R and L represent the size of the pattern gap in the tensile test. Smooth represents that there is no gap in the tensile sample. According to the obtained stress triaxiality and equivalent fracture strain.

The final J-C model is shown in Equation (4). The Oyane criterion is fitted with parameters to obtain the fracture criterion shown in Equation (5).

$$\sigma = \left(969.9 + 1012.3\varepsilon^{0.59}\right)\left(1 + 0.01ln\frac{\dot{\varepsilon}}{\dot{\varepsilon}_0}\right) \tag{4}$$

$$0.9 = \int_0^{\bar{\varepsilon}_f} (1 + 3.2\eta)d\bar{\varepsilon}_p \tag{5}$$

*2.2. Finite Element Simulation Model*

In this paper, a finite element model was established for simulation analysis. In order to explore the principle of force variation fine-blanking deformation, this paper selected

cylindrical parts for analysis. The diameter of the parts is 20 mm and the thickness is 4 mm. Since the part is axisymmetric, a 2D finite element model was used for analysis according to its structural characteristics [22]. The friction coefficient is determined from the experience of working with enterprises to do finite element simulation. The schematic diagram of the model is shown in Figure 2, and the parameters of the finite element model are shown in Table 3. In the fine-blanking process, the material often begins to fracture at the edge of the die, so the acquisition node of mechanical parameters was selected here.

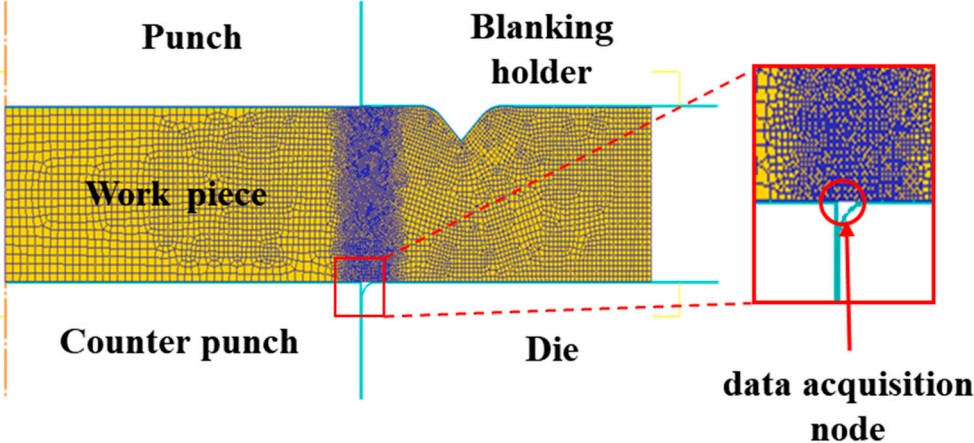

**Figure 2.** Schematic diagram of finite element model.

**Table 3.** Important parameters of the finite element simulation.

| Parameter | Value |
|---|---|
| Die diameter | 20 mm |
| Rounded radius of die | 0.5 mm |
| Punch diameter | 19.98 mm |
| Rounded radius of punch | 0.01 mm |
| V-ring height | 0.6 mm |
| Friction coefficient | $\mu = 0.1$ |
| Blanking speed | 5 mm/s |
| Shearing gap | 0.04 mm |

In the process of finite element simulation, the forming force changes mainly around the calculated value of the empirical formula. According to the empirical formula, the blank holder force is about 130 kN and the counter punch force is about 60 kN. In order to analyze the influence of different forming forces on fine blanking, two sets of simulation tests were designed. The forming force settings in the experiment are shown in Table 4.

**Table 4.** Finite element simulation design table.

| The First Group | | | The Second Group | | |
|---|---|---|---|---|---|
| Serial number | Blank holder force (kN) | Counter punch force (kN) | Serial number | Blank holder force (kN) | Counter punch force (kN) |
| 1 | 72 | | 6 | | 24 |
| 2 | 96 | | 7 | | 48 |
| 3 | 120 | 48 | 8 | 120 | 72 |
| 4 | 144 | | 9 | | 96 |
| 5 | 168 | | 10 | | 120 |

## 3. The Influence of Forming Force Change on Fine Blanking

Figure 3 shows the fracture surface characteristics of fine-blanking parts. Table 5 shows the effect of the forming force on the length of the clean cutting surface on the left.

The fracture surface of second group simulation is shown on the right. The proportion definition of the clean cutting surface is shown in Equation (6). It can be seen from these curves that the higher the forming force is, the longer the length of the clean cutting surface of the parts. Regardless of the blank holder force or counter punch force, when the values of them continue to increase during the loading process, the length of the clean cutting surface of the parts continues to increase. This shows that in the whole process of fine blanking, the forming force required to keep the material from fracture is not just a fixed value, but rather, it is in the process of constant change. In most of the process, it is gradually increased.

$$P_{cc} = 1 - \frac{L_{TB}}{t} \tag{6}$$

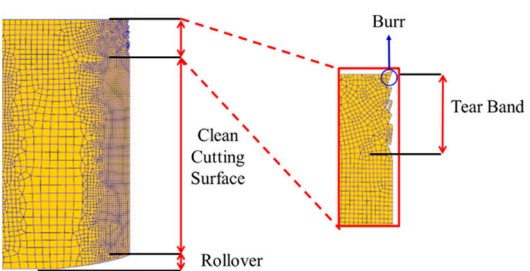

**Figure 3.** Typical quality characteristics of fracture surface.

**Table 5.** Finite element simulation results and fracture surface diagram.

| Change of Clean Cutting Surface of Fine-Blanking Parts When Forming Force Are Changed. | Number | Fracture Surface Diagram | $P_{CC}$ |
|---|---|---|---|
| | 1 | | 82% |
| | 2 | | 80% |
| | 3 | | 77% |
| | 4 | | 73% |
| | 5 | | 68% |

In Equation (6), $P_{cc}$ represents the proportion of the clean cutting surface, $L_{TB}$ represents the length of the tear band, and $t$ represents the thickness of the sheet metal. Since the roll-over is not the key research contents of this paper, the length of roll-over is also taken into account when calculating the proportion of the clean cutting surface.

At the same time, the improvement of forming force will bring higher load to the mold, as shown in Figure 4. As can be seen from the figure, the load on the mold is also increasing when the value of the forming force increases. Although the blank holder force and counter punch force are in a constant state in the setting, the load on the die is in a constantly changing state. From the trend, the mold load always increases first and then decreases. It can also be found that the peak load of the mold always appears in 20–40% of the whole blanking stroke, which is concentrated in the early stages of the whole stroke. The peak of mold load occurs within the area marked by the dotted box. At this time, although several sets of simulation forming force values are different, they can ensure that parts do not fracture. At the same time, the peak load of the mold is almost 25% reduced. This means that in the early and middle stages of blanking, appropriately reducing the forming force value can ensure the quality of parts while reducing the load of the die.

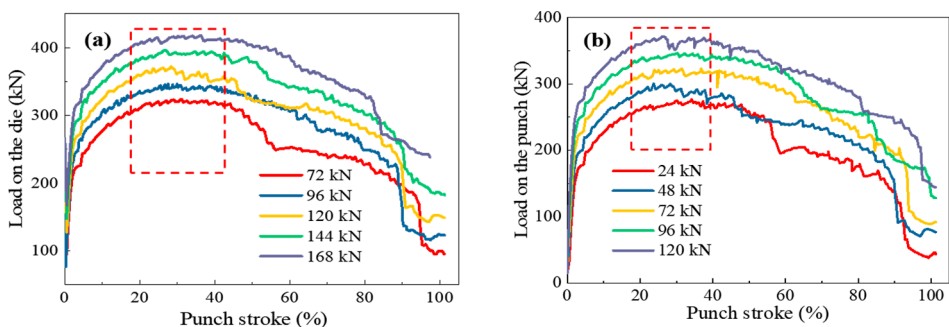

**Figure 4.** Influence of forming force changes on die load: (**a**) changes of die load when blank holder force is changed; (**b**) changes of die load when counter punch force is changed.

Taking the point marked with the red circle on the left side in Table 5 as an example, this point represents that the $P_{cc}$ of the part will reach 67% when the counter punch force is maintained at 48 kN and the blank holder force is maintained at 96 kN. Based on this, the length of the clean cutting surface can be determined by two forming forces. At the same time, the minimum value of the other forming force (blank holder force) can also be roughly determined when the proportion of the clean-cutting surface required by the part is given and the value of one forming force (such as counter punch force) is given. This value is the critical value of the blank holder force under the process conditions, and the state when the material is about to fracture can be considered as the critical state under the forming force combination.

According to this analysis, each data point in Table 5 can be regarded as the critical state of material fracture under the forming force. When the forming force is not enough to maintain the triaxial hydrostatic stress state of the material, the micro cracks expand and combine, and the material breaks obviously. This is the critical state of the material. Therefore, for the material in each stage of the blanking stroke, there is a risk of fracture, and there is a critical state of fracture. There is also a critical forming force combination in the critical state. When the forming force selection is better than the critical forming force combination in the loading process, the material will not fracture; otherwise, the material will fracture. In order to explore this thought, this paper carried out a finite element simulation according to the blank holder force loading curve shown in Figure 5. In this simulation, the blank holder force changes according to the loading curve, and the counter punch force remains unchanged at 48 kN.

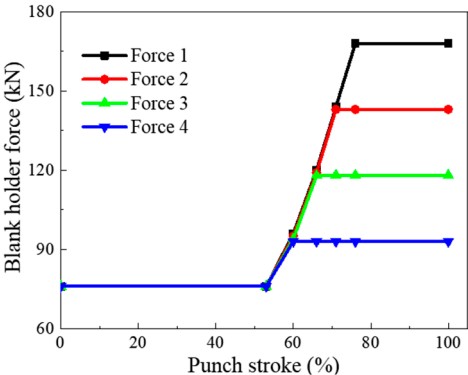

**Figure 5.** Load curve of blank holder force under force variation.

The loading curve of the blank holder force in Figure 5 is designed from Table 5. Take the curve of force 4 in Figure 5 and the points marked in Table 5 for example, the blank holder force in the blanking stroke less than 60% is designed to be smaller. (After a short blanking stroke, the material is about to break.) When the blanking stroke exceeds 60%, the blank holder force value reaches 96 kN and remains until the end of blanking. The rest of the curve in Figure 5 is generated this way. The simulation results are shown in Table 6. Thus, a force variation in the load forms a force-loading curve.

**Table 6.** Proportion of clean cutting surface contrast.

|  | Force 1 | Force 2 | Force 3 | Force 4 |
|---|---|---|---|---|
| Force variation loading | 80.4% | 76.6% | 71.0% | 65.4% |
| Constant force loading | 81.0% | 77.1% | 70.7% | 65.3% |

As shown in Table 6, the finite element simulation results are basically the same when the variable blank holder force was used for loading compared with the constant loading with high forming force. In the fine-blanking process, the same effect can be achieved without the need to maintain a high finishing force throughout the whole process.

Figure 6 shows the comparison of die loads under the two conditions of constant load and force variation load. In the simulation of constant load, the blank holder force was 168 kN, and the counter punch force was 48 kN. The results of variable load loading come from the simulation of force 1 in Figure 5, and the maximum blank holder force is also 168 kN. Combined with Table 6 and Figure 6, it can be found that in the simulation with force variation load, the peak load of the mold decreases by 25%, while the proportion of clean cutting surface remains almost unchanged. Therefore, force variation load can greatly reduce the load of the die while ensuring the size of the clean cutting surface.

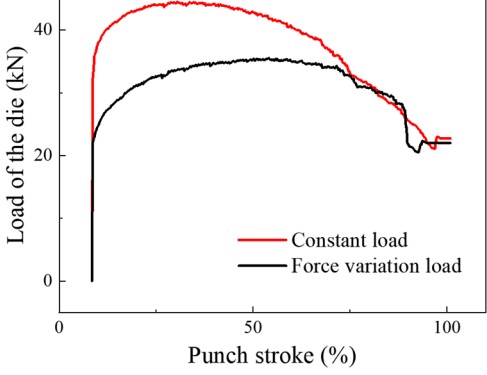

**Figure 6.** Comparison of mold load under force variation load and constant load.

Therefore, the force variation loading curve can be designed based on the forming force requirement of the workpiece and the characteristics of the die in the process of fine blanking. If there is a force variation loading curve, at every moment in the blanking stroke, the forming force can exactly meet the material for the three-way compressive stress state demand so that the material will not fracture; this curve is the optimal forming force loading curve. The application of this curve in the process of force variation load can minimize the unnecessary mold load caused by forming force. This will greatly reduce the damage to the die in the process of fine blanking of high-strength materials and reduce the difficulty of processing high-strength materials.

## 4. Neural Network and Genetic Algorithm Optimization Process

### 4.1. Neural Network Modeling

In the discussion of the previous chapter, a group of loading curves of force variation load were obtained by analyzing the existing forming force and clean cutting surface length data. However, there is some randomness in the process, and the final curve is not necessarily the optimal loading curve. Therefore, this section will find the optimal loading curve via an algorithm by using neural network and a multi-objective genetic algorithm.

In this paper, a BP neural network model was established through the MATLAB platform. The structure diagram of the neural network model is shown in Figure 7. The input layer of the neural network mainly involves two parameters. One is the blank holder force and the other is the counter punch force. In the process of modeling and optimization, it is necessary to find the optimal loading curve of the blank holder force and counter punch force, so both forming forces change with the blanking stroke. In the simulation, this paper divides the blanking process into three stages according to the time: the early stage, the middle stage, and the ending stage. Each stage is given a forming force value, and the schematic diagram of the final loading curve is shown in Figure 8. For high-strength and low-plastic materials, the proportion of clean cutting surface is generally difficult to exceed 90%. If the forming force continues to be regulated at this time, a great force will be required. Therefore, in the last 10% of the blanking stroke, the forming force will remain the same as the value of the previous stage. Each forming force value represents an input in the neural network. Therefore, although the input variable only involves two forming forces, there are six variables, and each variable represents a forming force value, as shown in Figure 8. The hidden layer has only one layer, the number of neurons is 12, and the number of neurons in the output layer is 1, which represents the length of the clean cutting surface [23].

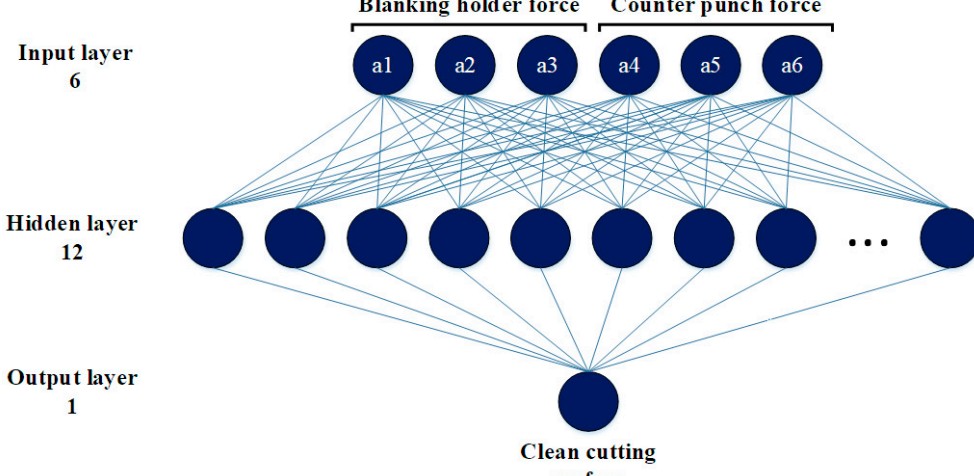

**Figure 7.** Neural network model structure of fine-blanking load process.

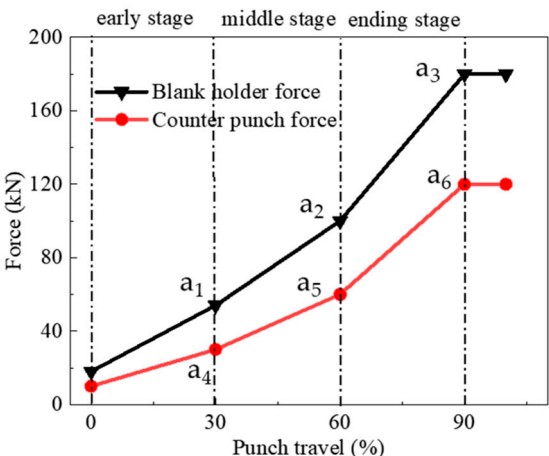

**Figure 8.** Schematic diagram of forming force variation loading.

In order to make the data of the neural network accurate enough, the neural network should be trained with a large number of samples before obtaining the relationship between the two forming force and the clean cutting surface of the fine-blanking part. As the punching stroke progresses, the forming force value should continue to increase, so the force variation finite element simulation is arranged according to the law. Part of the finite element simulation arrangement table is shown in Table 7.

**Table 7.** Finite element simulation arrangement of blanking under force variation (selected).

| Blanking Stroke Percentage | 0–30% | 30–60% | 60–90% | | 0–30% | 30–60% | 60–90% |
|---|---|---|---|---|---|---|---|
| | 0 | 0 | 0 | | 0 | 0 | 0 |
| | 180 | 180 | 180 | | 96 | 96 | 96 |
| | 40 | 180 | 180 | | 24 | 96 | 96 |
| | 40 | 40 | 40 | | 24 | 24 | 24 |
| | 40 | 40 | 100 | | 24 | 24 | 24 |
| | 40 | 40 | 180 | | 24 | 24 | 24 |
| | 40 | 100 | 100 | | 24 | 24 | 24 |
| | 40 | 100 | 180 | | 24 | 24 | 24 |
| | 40 | 180 | 180 | | 24 | 24 | 24 |
| Blank holder force (kN) | 100 | 100 | 100 | Counter punch force (kN) | 24 | 24 | 24 |
| | 100 | 100 | 180 | | 24 | 24 | 24 |
| | 100 | 180 | 180 | | 24 | 24 | 24 |
| | 180 | 180 | 180 | | 24 | 24 | 24 |
| | 40 | 40 | 40 | | 24 | 24 | 60 |
| | 40 | 40 | 100 | | 24 | 24 | 60 |
| | 40 | 40 | 180 | | 24 | 24 | 60 |
| | 40 | 100 | 100 | | 24 | 24 | 60 |
| | 40 | 100 | 180 | | 24 | 24 | 60 |
| | 40 | 180 | 180 | | 24 | 24 | 60 |

A total of 100 sets of finite element simulation data were set as the basic data for neural network modeling in this paper. At the same time, 75 sets of data were randomly selected from the basic data as the training set and 25 sets were randomly selected as the test set. After determining the structure of the BP neural network and analyzing the training results of it, it was finally determined that the learning function of the BP neural network adopts 'trainlm'. The final training result of the neural network is shown in Figure 9.

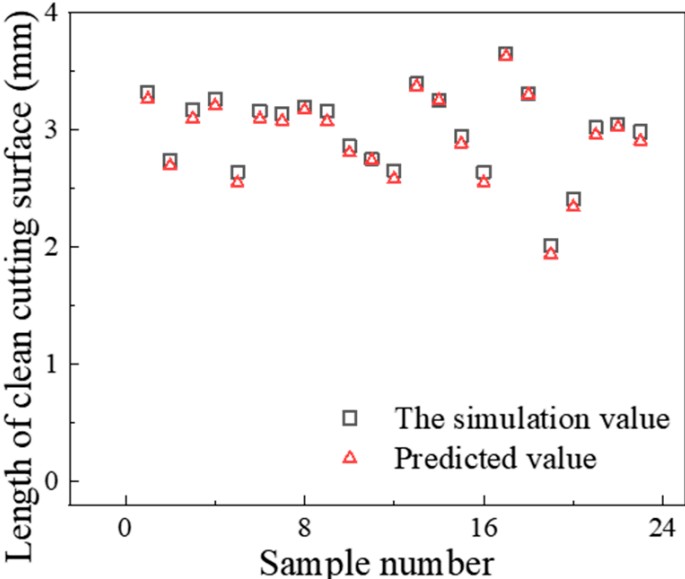

**Figure 9.** Comparison between network prediction results and finite element results.

As shown in Figure 9, it can be seen from the analysis of the prediction results of the neural network model that the prediction results of the neural network are very close to the results of the finite element simulation. Among them, the value of goodness of fit representing the accuracy of curve fitting reached 0.98, which proves that the accuracy of the neural network model is very high and can play a role in replacing the finite element simulation results.

### 4.2. Genetic Algorithm Optimization Process

In the process of optimizing, it is necessary to use multi-objective optimization to optimize the clean cutting surface of parts and manufacture cost. Different from the single-objective optimization problem, the sub-objectives in multi-objective optimization are often contradictory [24]. Obviously, in the process of optimizing, the lowest die load and the best quality for the cutting surface cannot be achieved at the same time. Therefore, in this process, we need to find the Pareto optimal solution for this problem.

First, the initial population is randomly coded in a given range in the form of real number coding. The initial population is designed to be 100, the number of iterations is designed to be 200, the crossover probability is 0.9, the mutation probability is 0.05, and the number of optimization targets is 2. The first optimization goal is the clean cutting surface length $S_1$ of the part, which represents the quality of the fine-blanking forming process. The other optimization goal is the sum of the numerical values of the two forming forces $S_2$ during the three stages, as shown in Equation (7). This optimization goal represents the die load and the damage to the mold during the fine-blanking process. The objective function value of each individual through the BP neural network is calculated and graded. Then, the non-inferior solution that meets the requirements is found and kept. If the set requirements are not met, continue with a series of genetic operator operations such as selection, crossover, mutation, etc. The feasible solutions that meet the conditions can be found finally until all Pareto solutions that meet the requirements are calculated. The final Pareto optimal solution is shown in Figure 10.

$$S_2 = a_1 + a_2 + a_3 + a_4 + a_5 + a_6 \qquad (7)$$

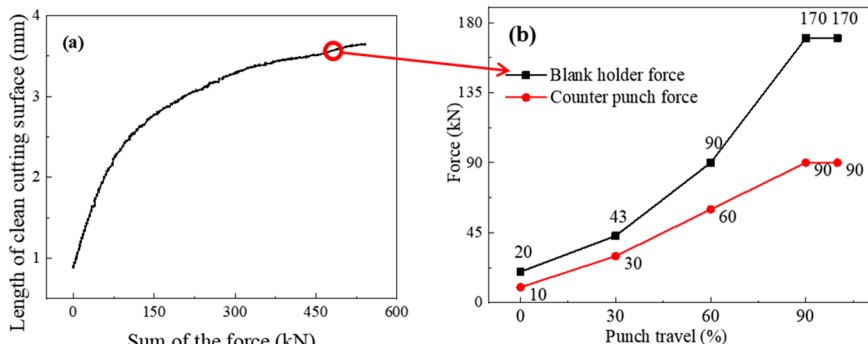

**Figure 10.** Pareto front and load route: (**a**) Pareto front; (**b**) the loading route represented by the points on the curve.

The Pareto optimal solution is a comprehensive result, and each point represents a set of loading curves composed of a blank holder force loading curve and counter punch force loading curve. However, in Figure 10a, it only shows the sum of the forming force. These combinations of curves are stored in the computed matrix, and it is difficult to show each set of curves in the Pareto front because it ultimately involves eight parameters. It can be seen from Figure 10 that the length of the clean cutting surface and the consumption of the forming force increase synergistically. At the same time, when the forming force increases to a certain extent, the growth of the length of the clean cutting surface has been extremely slow, and it is less efficient to continue to increase the forming force. Each data point obtained in Figure 10a represents the optimal solution under each clean cutting surface calculated by genetic algorithm.

Seven sets of data are selected from the Pareto optimal solution and compared with high forming force constant loading. Among them, the constant loading of maximum forming force means that the maximum forming force in the force variation process is used for simulation during the whole fine-blanking process. The constant loading of the average force means that the value obtained by averaging the sum of the forming forces in the force variation process maintains a constant loading during the entire stroke for simulations. This loading mode simulates the constant loading condition, which is approximately equal to the peak load of force variation load. The results in the table are calculated by inputting the loading paths in the table into the finite element simulation. The result comparison is shown in Table 8.

**Table 8.** Comparison between different loading modes.

| Loading Route | | | | | | | Result | |
|---|---|---|---|---|---|---|---|---|
| Blank holder force | | | Counter punch force | | | Force variation load | Maximum force constant load | Average force constant load |
| 0–30% | 30–60% | 60–90% | 0–30% | 30–60% | 60–90% | | | |
| 59.59 | 98 | 186.81 | 34.22 | 69.57 | 100 | 90.25% | 90.50% | 65.09% |
| 57.76 | 95 | 188 | 39.67 | 67.66 | 99.98 | 90.50% | 90.75% | 66.20% |
| 53.51 | 95.85 | 188 | 34.11 | 71.92 | 100 | 90.00% | 91.25% | 65.75% |
| 58.89 | 94.85 | 188 | 34.18 | 74.89 | 109.85 | 89.75% | 91.00% | 66.65% |
| 55.21 | 93.52 | 188 | 38.09 | 69.49 | 110 | 90.75% | 91.75% | 66.48% |
| 52.39 | 92.9 | 187.25 | 42.35 | 67.99 | 109.64 | 90.75% | 91.75% | 66.32% |
| 57.68 | 83.75 | 188 | 38.78 | 68.43 | 109.96 | 91.25% | 92.00% | 67.03% |

In Table 8, all loading curves in the table come from the Pareto front obtained above. The ratio of clear cutting surface in the table is the result of the finite element simulation with the previous curve. Except for the forming force, the other parameters in the simulation are consistent with the second section. The finite element simulation of the first line in Table 8 is sorted out. The results are shown in Figure 11 and Table 9. From Figure 11a, differences between the loading modes have been shown. Figure 11b shows the load curves of the die under different loading modes, and only the load changes of the die are shown here. Table 9 shows the fracture surfaces of parts under different loading modes. According to

Figure 11 and Table 9, it can be seen that the force variation load process can greatly reduce the load on the die on the premise of ensuring the quality of parts. High-quality parts can be obtained by loading with high forming force throughout the whole process, but the die is heavily loaded. If the forming force value is reduced, the load on the die will be reduced, but the quality of the parts will also become worse. The force variation load fine blanking can greatly reduce the load on the die on the premise of ensuring the quality of the parts. According to the calculation, in the loading mode of constant average force loading, the blank holder force is only 115 kN, but the peak load of the die is almost the same as the result of applying variable load loading. In the loading mode of variable load, the peak blank holder force reaches 180 kN. At the same time, it can also improve the quality of the parts under the condition that the peak load of the die remains unchanged.

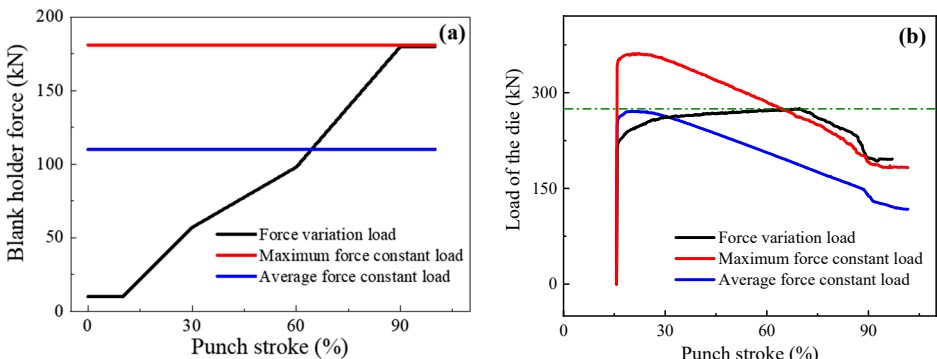

**Figure 11.** Diagram of loading path and load on the die: (**a**) loading route of blank holder force; (**b**) load on the die.

**Table 9.** Fracture surface quality of the parts.

| Loading Model | Fracture Surface Diagram | $P_{CC}$ |
|---|---|---|
| Force variation load | | 90.25% |
| Maximum force constant load | | 90.5% |
| Average force constant load | | 65% |

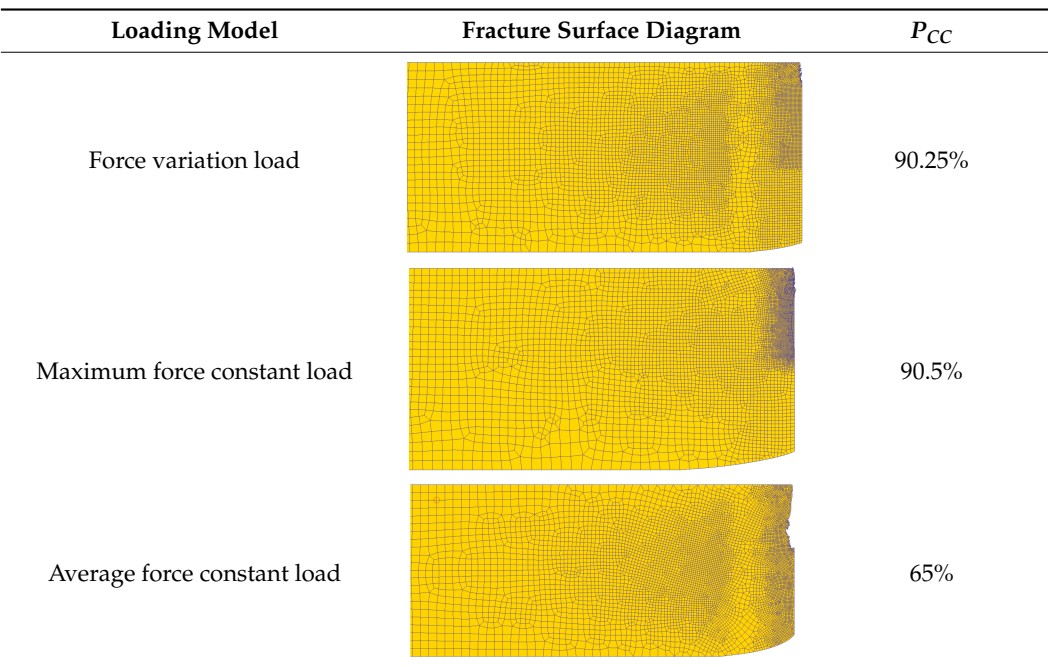

## 5. Results and Discussion

The criterion used in the analysis of fracture in this paper is Oyane criterion. In this criterion, there are three main factors affecting the fracture. They are equivalent stress, equivalent strain, and hydrostatic stress. This paper selects the equivalent stress and equivalent strain data in the simulation to conduct the analysis. The data acquisition node is selected at the edge of the die, as shown in Figure 2. Three sets of simulations with

obvious difference are shown in Table 10. Other parameters in the simulation are the same as those in the second section. From Figure 12, the numerical value and changing trend of the equivalent stress and the equivalent strain at the edge of the die in these simulations are almost the same. Only when the material has fractured obviously do the equivalent strain data in these simulations begin to be different. The equivalent stress and equivalent strain of the material during the entire forming process are only related to the properties of the material and the punch stroke. From the perspective of the mold structure, the blank holder force and the counter punch force cannot cause the parts to have obvious plastic deformation. The equivalent stress is directly related to the plastic state, so the equivalent stress and equivalent strain will not be significantly affected when the forming force changes in a certain range.

**Table 10.** Blank holder force and counter punch force in three groups of simulations.

|  | Blank Holder Force (kN) | Counter Punch Force (kN) |
|---|---|---|
| Process 3 | 150 | 48 |
| Process 2 | 180 | 72 |
| Process 1 | 120 | 96 |

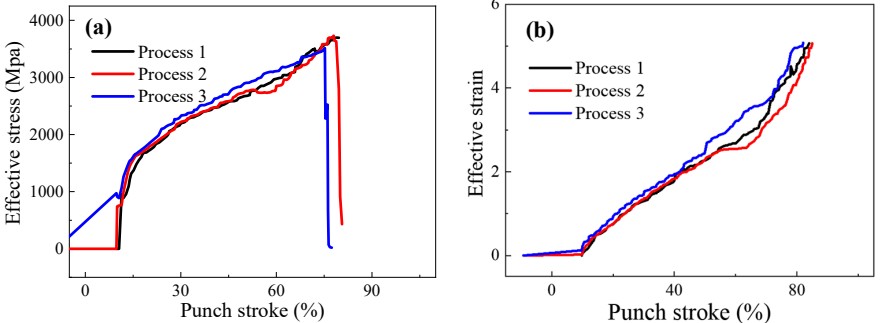

**Figure 12.** Equivalent stress and strain data of three sets of simulations: (**a**) the equivalent stress changes; (**b**) the equivalent strain changes.

Figure 13a is the stress diagram of the material in the process of fine blanking. A hexahedron element is arbitrarily selected in the shear deformation zone inside the material, and the stress distribution is shown in Figure 13b. $P_y$ is the resultant force of punch and counter punch acting on the material. $P_v$ is the force exerted by the V-ring on the material; N is the lateral force of the die acting on the material; $F_x$, $F_y$ represent the friction between the material and the die, and t is the thickness of the sheet. In Figure 13b, all normal and shear stresses are generated by the corresponding forces given in Figure 13a. $\sigma_z$ is the normal stress caused by the constraint of the die on the material.

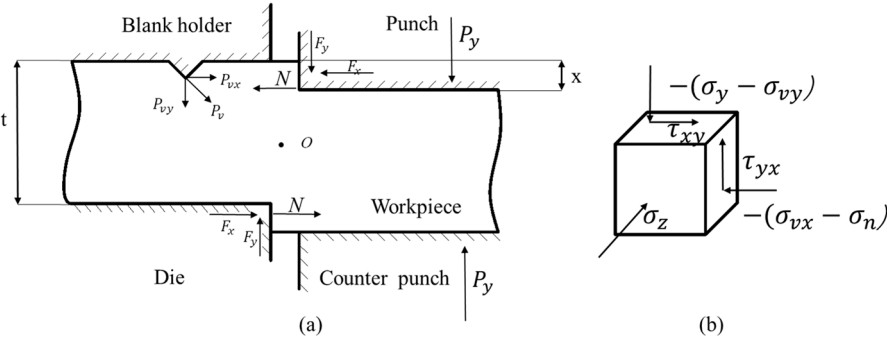

**Figure 13.** Schematic diagram of stress on the fine-blanking material and material in the shear deformation zone: (**a**) stress diagram of the material in the process of fine blanking; (**b**) stress distribution of the material in the deformation zone.

When the forming force changes, the equivalent stress and strain of the material do not change. Therefore, in the fine-blanking process, the only way to delay the fracture is to regulate the hydrostatic stress, as shown in Equation (8). The stress tensor $T_\sigma$ at a point in the material can be uniformly abbreviated as:

$$T_\sigma = T_\sigma' + T_\sigma'' \tag{8}$$

where $T_\sigma'$ is the stress sphere tensor, which produces elastic deformation; $T_\sigma''$ is the partial tensor of stress and produces plastic deformation. Equation (9) is obtained by derivation.

$$
T_\sigma =
\begin{pmatrix}
-(\sigma_{vx}+\sigma_n) & \tau_{xy} & 0 \\
\tau_{yx} & -(\sigma_{vy}+\sigma_y) & 0 \\
0 & 0 & -\sigma_z
\end{pmatrix}
= T_\sigma' + T_\sigma'' =
\begin{pmatrix}
-\sigma_m & 0 & 0 \\
0 & -\sigma_m & 0 \\
0 & 0 & -\sigma_m
\end{pmatrix} +
$$
$$
\begin{pmatrix}
-\frac{2}{3}(\sigma_{vx}+\sigma_n)+\frac{1}{3}(\sigma_y+\sigma_{vy}+\sigma_z) & \tau_{xy} & 0 \\
\tau_{yx} & \frac{1}{3}(\sigma_{vx}+\sigma_n+\sigma_z)-\frac{2}{3}(\sigma_y+\sigma_{vy}) & 0 \\
0 & 0 & -\frac{2}{3}\sigma_z+\frac{1}{3}(\sigma_{vx}+\sigma_n+\sigma_y+\sigma_{vy})
\end{pmatrix} \tag{9}
$$

The stress sphere tensor can be deduced into the form of average stress, as shown in Equation (10).

$$-\sigma_m = \frac{1}{3}\left(\sigma_{vx}+\sigma_{vy}+\sigma_n+\sigma_y+\sigma_z\right) \tag{10}$$

It can be seen from Equation (10) that the hydrostatic stress of material in the shear deformation zone is related to the blank holder force, the counter punch force, and the constraints of the mold. In the early stage of fine blanking, the deformation of the material is almost all concentrated in a narrow area, so the hydrostatic stress state can be maintained without high forming force. Although in several parts of the process, the forming force value of the difference is very large, in the early and middle blanking stroke, under the restriction of the die, the hydrostatic stress value still remains negative. At this time, the material deformation degree is also small, and the stress triaxiality increases little. The cavity-type damage inside the material is not obvious. In the middle and ending stage of fine blanking, the ratio of blanking clearance and die clearance to the thickness of the sheet not yet blanked continues to expand. At this time, the area of tensile stress in the deformation zone continues to expand, and the micro holes and micro cracks continue to expand. These conditions lead to a growing demand for forming force. As shown in Figure 14, the variation trend of hydrostatic stress and damage value is consistent with that mentioned above.

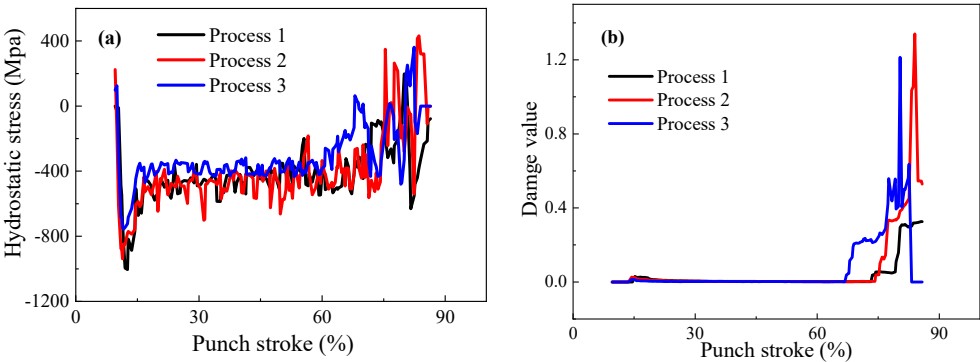

**Figure 14.** Hydrostatic stress and damage value data of three groups of simulation: (**a**) hydrostatic stress; (**b**) damage value.

The change trend of the load on the die in the blanking stroke is different. Taking the die as an example, the load on the die mainly comes from two aspects, the blank holder

force $F_{bh}$ and the pressure of material plastic deformation on the die during blanking. $F_p$ in the blanking process, as shown in Equation (11).

$$F_{die} = F_p + F_{bh} \tag{11}$$

Among them, in the conventional fine blanking, the counter punch force $F_{bh}$ remains constant. The material to be processed by fine blanking gradually decreases with the blanking stroke, so $F_p$ gradually decreases with the blanking stroke, and its peak appears in the early stage of the blanking stroke. This leads to a gradual decrease in the load of the die during the conventional fine-blanking process. The peak stage of the load on the mold does not coincide with the peak stage of the material demand for forming force. Therefore, if the trend of material demand for forming force is fully considered in the design of the force variation load curve, the load of the mold can be greatly reduced on the premise of ensuring the quality of parts, as shown in Figure 15. The deformation force in Figure 15 is $F_p$; it represents the pressure on the die when the material is plastic deformed during the blanking stroke. Of course, because the mold load can be effectively reduced in force variation load fine blanking, when designing the loading curve, appropriately increasing the forming force can achieve the effect of improving the clean cutting surface of parts, while the mold load remains unchanged or even reduced.

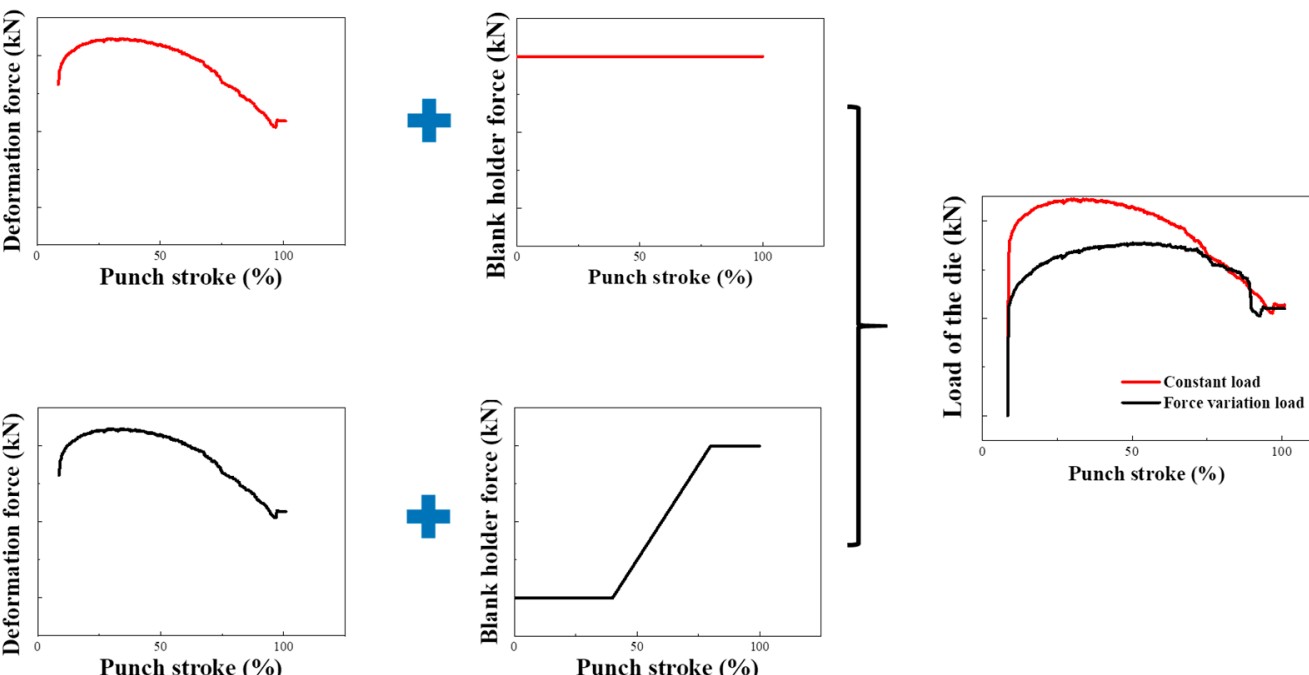

**Figure 15.** Difference of mold load between variable load and constant load.

The 2D models used in the above finite element simulation are all cylinders. According to the empirical formula, factors such as material, sheet thickness, and part size should be considered when calculating the forming force [25], as shown in Equations (12)–(14). Therefore, if the optimization model proposed in this paper is applied to different models and parts, the forming force under variable load should be adjusted according to the size and shape of parts in proportion.

Empirical formula for main blanking force:

$$F_p = L \cdot t \cdot \sigma_b \cdot f_1 \tag{12}$$

Empirical formula for blank holder force:

$$F_{bh} = L_R \cdot 2h \cdot \sigma_b \cdot f_2 \tag{13}$$

Empirical formula for counter punch force:

$$F_{cp} = 20\% \cdot F_p \tag{14}$$

In these formulas, $F_{bh}$ represents the blank holder force, $F_{cp}$ represents the counter punch force, and $F_p$ represents the main blanking force. $L$ represents the shear line length of the part, $L_R$ represents the length of the gear ring, $t$ represents the thickness of the part, and $\sigma_b$ represents the tensile strength of the material of the part. $f_1$ and $f_2$ respectively represent the coefficients related to the tensile strength of the material.

For example, $A$ and $B$ are two parts with the same material but different shapes, and their blank holder force can be transformed by Equation (15).

$$\frac{F_{bh\ A}}{F_{bh\ B}} = \frac{L_{R\ A} \cdot h_A}{L_{R\ B} \cdot h_B} \tag{15}$$

The prediction of the forming force in this paper is based on the deformation law of the material. Therefore, when changing different materials, the forming law does not change linearly with the tensile strength of the material, so it may not be applicable.

In order to verify the effect on the length of the clean cutting surface by force variation fine blanking, experiments were carried out on a hydraulic servo fine-blanking press, in which the blank holder force and counter punch force can be servo controlled. In order to determine whether the optimized model finally obtained in this paper can be applied to other shapes of parts, experiments were first carried out on parts with simple shapes. The material for the experiment is TC4 alloy, and the part thickness is 4 mm. The parts fabricated by the force variation loading are compared with those processed by the average force constant loading.

The thickness of the parts is 4 mm, and the processing goal of the parts is that the proportion of the clean cutting surface reaches 90%. According to the proportion of clean cutting surface, the optimal forming force loading curve can be found in the optimized model. The forming force in the loading curve can be transformed according to the previous formula to obtain the optimal forming force loading curve for the target part. The loading condition of force variation the blank holder force is 81 kN in the early stage, 165 kN in the middle stage, and 223 kN in the late stage; the counter punch force is 42 kN in the early stage, 68 kN in the middle stage, and 110 kN in the late stage. For the loading condition of the average force constant load, all forming force values are calculated as mentioned before. The blank holder force and counter punch force for this part are 190 kN and 80 kN, respectively. It is guaranteed that the two die loads are close to each other under the two loading modes.

The comparison chart is shown in Figure 16. Figure 16a,c show the upper surface and fracture surface of the parts produced by force variation fine blanking. Figure 16b,d show the upper surface and fracture surface of the part processed by conventional fine blanking. When the parts are processed by the force variation fine blanking, the length of the clean cutting surfaces is about 3.6 mm. The proportion of clean cutting surface reached over 90%. It is very close to the original forming target. However, the parts processed by the constant loading of the average force have poor cutting surface quality. It is only 2 mm in length, and it is quickly turned into a fracture zone. This proves that force variation load fine blanking can improve the quality of parts under the same mold load. Experiments show that when the shape of the part is relatively simple and the material does not change, the prediction accuracy of the model is very high, and the optimization effect of the part is obvious.

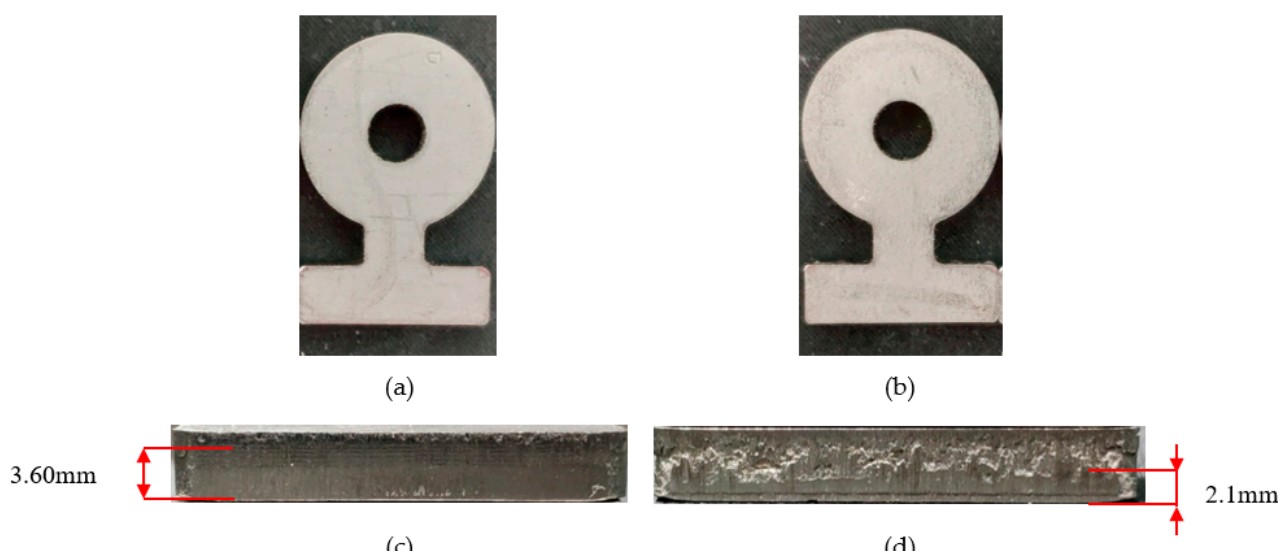

**Figure 16.** Comparison between the process of constant average force load and force variation load;
(**a**) top surface of parts produced by force variation load applied; (**b**) top surface of parts produced by average force constant load applied; (**c**) fracture surface of parts produced by force variation load applied; (**d**) fracture surface of parts produced by average force constant load applied.

## 6. Conclusions

(1)   The variation trend of equivalent stress and equivalent strain is only affected by the material and blanking stroke. It is determined that the forming force mainly influences the length of the clean cutting surface by influencing the hydrostatic stress.

(2)   The forming force demand of the workpiece increases gradually with the blanking stroke, and its peak value is in the late blanking stroke. The peak value of the load on the die decreases gradually with the blanking stroke after the punch contacts the material, and its peak value is in the early blanking stroke. The trends of the two of them are opposite.

(3)   This difference between them makes it possible for an appropriate force variation load that forms a force-loading curve to greatly reduce the mold load while ensuring the quality of parts, or to greatly improve the quality of parts while keeping the mold loads close to another.

(4)   The optimal processing curve of the force variation fine-blanking process was obtained through neural network and genetic algorithm. The optimal loading curve of the forming force should be kept at a low level in the early stage of the blanking stroke, and it should increase gradually in the middle and ending stage.

Force variation load fine blanking can effectively improve the quality of fine-blanking parts while keeping the mold load unchanged. This will effectively reduce the wear of the mold in the production process and the manufacturing cost of high-strength materials. It is conducive to the popularization and application of high-strength materials. So far, the experiment has only been performed on simple shape parts, and the application experiment of the model on more complex shape parts has not been completed. The application of variable load needs a servo fine-blanking press; it is difficult to use an ordinary fine-blanking machine for a variable load process to achieve the mass production of parts. In the future, we will calculate the loading curve of variable load for the fine-blanking process through theoretical levels and propose the theoretical model or empirical formula of the variable load processing curve for different materials, thicknesses, shapes, and processing requirements.

**Author Contributions:** H.C. was responsible for the reading and induction of all literature; Y.L. and H.M. were in charge of the whole trial; K.J. was engaged in literature collection. All authors read and approved the final manuscript.

**Funding:** Support from the National Key R&D Program of China (No. 2020YFA0714900).

**Institutional Review Board Statement:** Not applicable.

**Informed Consent Statement:** Not applicable.

**Data Availability Statement:** The raw/processed data required to reproduce these findings cannot be shared at this time as the data also forms part of an ongoing study.

**Acknowledgments:** The authors are grateful for the financial support from the National Key R&D.

**Conflicts of Interest:** The authors declare no competing financial interest.

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
