# Peer review of "A Novel Force Variation Fine-Blanking Process for the High-Strength and Low-Plasticity Material"

_metals, doi:10.3390/met12030458_

Round 1

Reviewer 1 Report

The paper is clear and interesting. Before the publications are suggested the following minor revisions:

  1. Rearrange the abstract, in the present form it is too long and not focalized on the aim of the work.
  2. Explain better the tensile tests equipment and their set up. In addition, comments about Fig.1 and its legend are need;
  3. Check the width of columns 1 and 4 of Table 2;
  4. Explain better how the burnish zone extent was evaluated;
  5. In the y-axes of figures, indicate the unit of measurement between parentheses and not after /.
  6. In general, the results and discussion section needs to be rearranged. Some results are reported in section 2. Check this.

Reviewer 2 Report

The paper concerns fine blanking process, especially high-strength materials with low plasticity. To improve the quality of the cut surface the Authors combine genetic algorithm and neural network methods to find the optimal cutting force (blank holder force and counterpunch force). The paper can be considered for publication in Metals, however, before acceptance, it's necessary to improve:

  • the quality of writing; the manuscript must be checked by a native speaker; sometimes it is difficult to understand the authors' intentions
  • Introduction: add more current references (2020 – 2021). 60% references are from 2000-2015; no reference to the achievements of the last 2 years
  • add the chemical composition for TC4 titanium alloy and its mechanical properties
  • Fig.1b: how was stress triaxiality defined? Unit? If dimensionless, make it clearly [-] . Explain R and L ?
  • On what basis was the friction coefficient 0.1. Is this the coefficient of friction between titanium and steel under dry friction conditions?
  • Fig.3: how was burnish zone ratio defined?
  • Add figure presenting the appearance of blanked surface with regions: roll-over, burnish, fracture and burr
  • Have you verified the results experimentally? How is a variable force realized in practice?
  • Fig.14. What is for constant average force load and what for force variation load

Reviewer 3 Report

In this manuscript, a novel force variation fine blanking process was proposed to improve the part quality of high-strength and low-plasticity material. The authors showed the use of force variation control in fine blanking process and claimed it as an novelty. This is an originality in terms of application and it must set the optimal force variation case by case. But, in terms of scientific soundness, this manuscript is still weak. The authors should find out the scientific content and clarify it clearly. Note, more information about the FEM simulation and experimental verification should be provided.

Reviewer 4 Report

In the abstract, the authors of the article use the terms high accuracy and good surface quality. It is important for the user (a specialist in the field of fine blanking) to know in which accuracy class IT or what roughness Ra, resp. what is the length of clean cutting surface of parts is achieved by this technology. This is an error - abstract does not appeal to the reader.

The reader of the article should find out when reading the abstract what research methods were used, what material was used and what effect resp. what agreement with the actual results is obtained using the above techniques and how the results obtained were verified, i. experiment or in the production of real components.

In the introduction, the authors conducted a good search of knowledge on the issue of fine blanking. In the introduction to the article it is necessary to add the scheme of the fine blanking process, the state of stress in the material cutting zone and state, forces, shear gap, how thick materials can produce parts by fine blanking, using a holder with a pressure edge in the form V groove on one side sheet metal or on both sides of fine sheet metal and other limitations of this technology.

At the end of the introductory chapter, it is necessary to clearly define the goal of the research. For example: The aim was to determine the effect of the holding force on the length of the smooth surface in the production of parts of a given shape.

In Chapter 2, it is necessary to add describe the meaning of individual symbols in Equations 1 and 2, how the individual constants were determined and in which SI units it is necessary to enter them into the used simulation software.

In this chapter it is necessary to add what material was used. In the Deform program, it is enough to define the process of fine blanking cutting with the data given in tab. 1?  Add what was the shearing gap (its value), as was determined the coefficient of friction? At what interval were the strain rate values ​​changed, which material data are needed, and at what interval were they changed? What equipment was used to determine the values ​​of mechanical properties, what dimensions of samples were used, number of samples, etc. What is the assumed accuracy of the shear force results, the clean cutting surface of parts length obtained by the simulation, so that we can state that the neural network algorithm used is able to predict the shear edge length with the specified tolerance and so on? It should be clearly stated that the results were obtained by numerical simulation in the program Deform for neural network training and verification of the algorithm used. It is also necessary to add for which shapes a neural network can be used. Or apply general to all shapes.

Figure 10 shows a comparison of the results obtained by the simulation and calculated by the generic algorithm. However, the reader (user) does not obtain information about the effect of the force of the blank holder and the effect of the counter punch force.

In Table 5, clearly indicate which results were obtained by simulation and which by prediction.

The name of Table 6 does not describe the content, because the table shows the holding force and the ejector force. What does the term deformation force mean - fig 13?

In the figures and tables, the SI units must be given in brackets as indicated in the tables.

The authors present the application of neural networks for the prediction of selected indicators of fine blanking. The topic is current, it is a topic for the very near future.  Finally, the authors should add (specify) what is the scientific contribution of their article. What are the limitations of the use of the obtained results, what do they plan in the future?

Round 2

Reviewer 2 Report

Thank you for the corrections made. I find them satisfying

Reviewer 3 Report

Thank you for the authors responses. These are not clearly shown Originality / Novelty, Significance of Content, and Scientific Soundness.

Reviewer 4 Report

Dear authors,
you have adequately incorporated my comments. I am satisfied.